# Optimal Disease Surveillance with Graph-Based Active Learning

JOSEPH L.-H. TSUI*, Department of Biology, University of Oxford, UK

MENGYAN ZHANG*, Department of Computer Science, University of Oxford, UK

PRATHYUSH SAMBATURU, Department of Biology, University of Oxford, UK

SIMON BUSCH-MORENO, Department of Biology, University of Oxford, UK

OLIVER G. PYBUS, Department of Biology, University of Oxford, UK, Department of Pathobiology & Population Science, Royal Veterinary College, UK, and Pandemic Sciences Institute, University of Oxford, UK

SETH FLAXMAN†, Department of Computer Science, University of Oxford, UK

ELIZAVETA SEMENOVA†, Department of Epidemiology and Biostatistics, Imperial College London, UK

MORITZ U. G. KRAEMER†, Department of Biology, University of Oxford, UK and Pandemic Sciences Institute, University of Oxford, UK

The detection and tracking of the spread of emerging pathogens is critical to the design of effective public health responses. Policymakers face the challenge of allocating finite testing resources across locations, with the goal of maximising the information obtained about the underlying disease distribution. We model this decision-making process as an iterative node classification problem on an undirected and unweighted graph, in which nodes represent locations and edges represent movement of infectious agents among them. To begin, a single node is randomly selected for testing and determined to be either infected or uninfected. Test feedback is then used to update estimates of the probability of unobserved nodes being infected and to inform the selection of nodes for further testing at the next iteration, until a certain resource budget is exhausted. Under this framework, we evaluate and compare the performance of previously developed Active Learning policies, including node-entropy and Bayesian Active Learning by Disagreement. Using data from simulated outbreaks on both random and empirical human mobility networks, we explore the performance of these policies under different outbreak scenarios and graph structures. Further, we propose a novel policy that considers the distance-weighted average entropy of infection predictions among the unobserved neighbours of each candidate node. Our proposed policy outperforms existing ones in most outbreak scenarios, leading to a reduction in the number of tests required to achieve a certain predictive accuracy. Our findings could help design cost-effective surveillance policy for emerging and endemic pathogens, accelerating disease detection in resource-constrained situations.

---

*Both authors contributed equally to this research.

†Authors jointly supervised this work.

Authors' Contact Information: Joseph L.-H. Tsui, lok.tsui@new.ox.ac.uk, Department of Biology, University of Oxford, Oxford, UK; Mengyan Zhang, Department of Computer Science, University of Oxford, Oxford, UK, mengyan.zhang@cs.ox.ac.uk; Prathyush Sambaturu, Department of Biology, University of Oxford, Oxford, UK; Simon Busch-Moreno, Department of Biology, University of Oxford, Oxford, UK; Oliver G. Pybus, Department of Biology, University of Oxford, Oxford, UK and Department of Pathobiology & Population Science, Royal Veterinary College, London, UK and Pandemic Sciences Institute, University of Oxford, Oxford, UK; Seth Flaxman, Department of Computer Science, University of Oxford, Oxford, UK; Elizaveta Semenova, Department of Epidemiology and Biostatistics, Imperial College London, London, UK; Moritz U. G. Kraemer, Department of Biology, University of Oxford, Oxford, UK and Pandemic Sciences Institute, University of Oxford, Oxford, UK.

CCS Concepts: • **Applied computing** → *Health care information systems*; *Multi-criterion optimization and decision-making*.

Additional Key Words and Phrases: Network Dynamics, Active Learning, Disease Surveillance

## 1 INTRODUCTION

Disease surveillance is critical for managing infectious disease outbreaks, as it enables public health authorities to monitor and respond to ongoing disease spread. Notable examples in the past decade include the 2014-2016 West African and 2018-2020 Kivo Ebola epidemic, and more recently, the COVID-19 pandemic, where early detection of the virus and continued tracking of its spread helped inform the design of effective interventions including targeted vaccinations [20, 23, 26, 45, 46], case isolation [2, 10, 12, 24, 27] and social distancing [6, 13, 15, 18]. Without timely and accurate surveillance data, the effectiveness of these interventions would likely have been compromised, with potentially increased public health risks and greater socio-economic disruptions. For example, it has been shown that travel restrictions targeted at countries where new variants of SARS-CoV-2 were first observed were rendered largely ineffective by delay in case detection and insufficient pathogen sequencing [42, 44]. Similarly, the lack of baseline testing prior to the 2015-2016 Zika epidemic likely contributed to the delay in the identification of the scale of disease spread, thereby allowing the virus to propagate to numerous countries before a global response was initiated [16, 21].

Well documented examples of effective disease surveillance have been largely limited to within-country efforts (e.g., the Real-time Assessment of Community Transmission (REACT) in the UK [36], and the National Notifiable Diseases Surveillance System (NNDSS) in the US [14]), while globally coordinated programs remain rare [33]. This leads to disproportionate and inequitable distribution of testing resources both within and between regions or countries, with some local authorities able to conduct large-scale mass testing for sustained periods of time, while others manage only sparse and sporadic testing [19, 49]. One study showed that the intensity of genomic sequencing during the COVID-19 pandemic was positively associated with Research & Development expenditures at a

country-level [8]. This likely allowed the virus to continue proliferating undetected in locations with insufficient testing, potentially prolonging local outbreaks.

## 1.1 Related Work

Previous research on disease surveillance has primarily focused on developing models to identify sentinel sites, with the objective of classifying nodes in networks that could serve as observational units for monitoring disease spread [1, 3, 34]. Recently since the COVID-19 pandemic, there has been a growing interest in the design of optimal control measures to contain disease spread [4], with some studies examining the cost-effectiveness of different strategies for testing and isolation in reducing transmission intensity; one recent study also explored the impact of different air travel regulations on the likelihood of a local epidemic escalating into a global pandemic [40]. However, the effectiveness of these interventions ultimately depends on the capacity of local authorities to conduct disease surveillance and to collectively provide an accurate assessment of the overall disease distribution at any stage of an outbreak - a challenge which, to the best of our knowledge, has received little attention to date [47].

Our work in this study attempts to fill this research gap by formulating the problem of disease surveillance as a node classification task with Active Learning (AL). Active node classification is a well-studied problem and a comprehensive comparison of the performance of existing AL methods on empirical graphs is presented in a recent study [29]. More recent development in this area has primarily focused on the design of methods that incorporate node attributes, particularly with applications on large Graph Neural Networks (GNNs) [9, 30, 48], as a result of recent advances in GNNs and the use of increasingly large datasets for model training. There has been limited work to date, however, on the use of these methods in the context of disease surveillance, or epidemiology, in general.

## 1.2 Contributions

Our contributions in this study are summarised below:

- We formulate the problem of designing an appropriate policy as a node classification problem with Active Learning on an undirected and unweighted network, where nodes represent locations and edges represent movement of infectious agents between locations. We allocate tests to a selected node at each iteration via a policy with the goal of achieving the best possible classification performance with a given budget.
- We design an adaptive test deployment framework to evaluate and compare the performance of different allocation policies in the context of disease surveillance, as shown in Figure 1.
- We propose a novel policy, named *Selection by Local-Entropy (LE)*, which takes into consideration graph-based uncertainties in its decision-making. We evaluate the performance of our proposed policy alongside existing AL policies (Table 1) under various outbreak scenarios and on networks with different structural properties, including those commonly found in empirical human mobility networks.
- We show that the performance of a given policy depends on both the test budget available and the geometry of the

underlying disease distribution, which is in turn determined by the network structure and stage of outbreak progression.
- Our proposed policy outperforms existing uncertainty-based policies in most scenarios, highlighting a need to consider the trade-off between exploration (sampling of unlabelled regions) and exploitation (sampling of regions with known heterogeneous disease distribution).

## 2 BACKGROUND AND PROBLEM SETUP

### 2.1 Disease Surveillance as a Node Classification Task

We consider the deployment of a disease surveillance program on a mobility network as a node classification task, where the goal of a policymaker (or agent) is to predict the presence or absence of a disease of interest (or whether the disease prevalence is below or above a certain threshold) at any unobserved node, provided that the infection status of a subset of nodes are known. We assume that the mobility network can be represented as an undirected and unweighted graph $G = (V, E)$ with $|V| = N$ such that each node $v_i \in V$ represents a location, and an edge $(v_i, v_j) \in E$ indicates the existence of movement of infectious agents between the nodes $v_i$ and $v_j$. Prior to the start of a surveillance program, it is assumed that there is an underlying disease distribution resulting from an outbreak that originated from a single infected node. Importantly, we assume that i) the outbreak can be modelled as a stochastic Susceptible-Infected (SI) process, where transmission can only occur between an infected node and an uninfected node if there is an edge between them (see Appendix A. of the supplementary document [43]), and ii) that the timescale over which transmission occurs is sufficiently longer than the timescale over which testing is deployed, such that the underlying disease distribution can be considered to be static over the course of the surveillance program. To indicate the underlying disease distribution, each node $v_i$ in the mobility network is assigned a binary label $y_i \in \{0, 1\}$ representing its infection status, where $y_i = 1$ if the node is infected (presence of disease of interest) and $y_i = 0$ if uninfected (absence of disease of interest). Exploration of the impact of relaxing these assumptions will be addressed in future work.

In a typical outbreak scenario, there is often little information available to inform the initial allocation of testing resources; it is also possible for the initial observation to be of either infection status under certain contexts of disease surveillance (e.g., during baseline monitoring of an endemic disease among wildlife reservoirs). With these in mind, here we assume that each surveillance program begins with the known infection status of a single (randomly selected) node. Given this initial observation, a hypothetical policymaker is then tasked with answering the question: **how should a finite amount of testing resources be deployed across the network to maximise the information gained about the underlying disease distribution?**

### 2.2 Test Allocation as an Active Learning Task

The study of this question is known as Active Learning (AL) [39], where the objective is to maximise the predictive performance of a model in training (also known as a surrogate model) with the

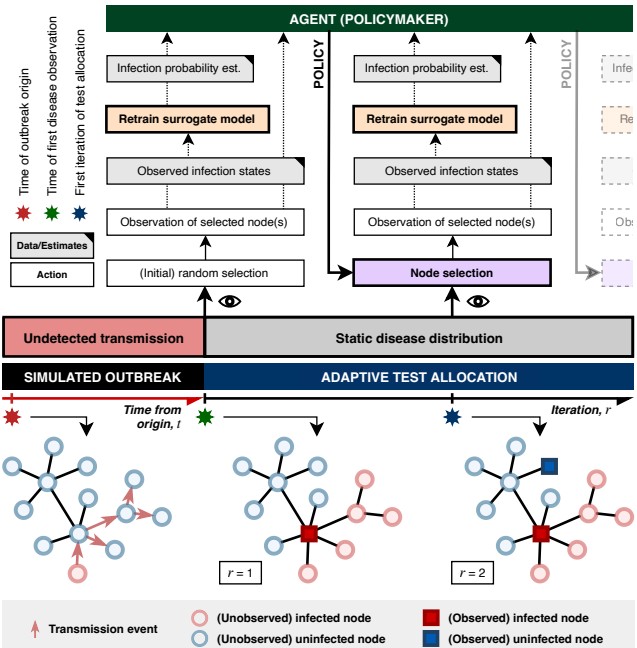

Fig. 1. A schematic illustration of the adaptive test deployment framework. The flow of information or data (grey boxes with a black corner) from one component to another is represented as arrows. The eye symbol indicates when the true infection status of a selected node is queried from the underlying disease distribution at each test iteration. Note that the simulated outbreak terminates at the time of first observation, i.e. the underlying disease distribution is assumed to be static over the course of test deployment.

fewest instances of labelled data possible. Since the process of labelling data can be expensive and time-consuming, the selection of data instances for labelling can alternatively be performed in an iterative fashion, where a small number of unlabelled data instances are selected at each iteration by an AL policy. The labels of these selected data instances are then revealed and used as input to retrain the surrogate model and to generate label predictions for unlabelled instances. Note that the label of each node is assumed to be unchanging between iterations, following the assumption of static disease distributions as described earlier.

A number of previously developed approaches exist for constructing the surrogate model, e.g., label propagation [50], Gaussian Random Field [51], and more recently, GNNs [25]. Here we adopt an approach that is particularly popular in spatial epidemiology, known as the Conditional Autoregressive (CAR) model, which assumes that the value of a variable at a given node in a network is conditional on the values at neighbouring nodes, with weights specified by the adjacency matrix, $\mathbf{A}$. In the context of disease surveillance with binary infection status, the CAR model allows us to estimate $p(v_i|\mathbf{D_r})$ for a given node $v_i$, i.e. the probability that the node is infected conditioned on the observed data $\mathbf{D_r} = \{(v_j, y_j)|\forall v_j \in V_{obs,r}\}$, where $y_j$ is the observed infection status of a node $v_j$, with $V_{obs,r}$ being the set of nodes with known infection status up to the current iteration $r$ (see 3.1 for a more detailed description). Note that it is not the focus

of this study to consider the predictive performance of different surrogate models, but rather the relative performance of different test allocation strategies given the same surrogate model. In future extensions where additional model complexities are incorporated (e.g., weighted and directed edges, node attributes), the performance of the different strategies given different surrogate models should be compared and assessed.

At each iteration, the predicted infection status generated by the surrogate model are used to guide the selection process at the next iteration, according to the acquisition policy of choice. One important group of AL policies is uncertainty-based, i.e. selecting nodes for observation according to where the surrogate model's predictions are maximally uncertain. One common measure of uncertainty is the information entropy of label predictions - the larger the entropy, the more uncertain the model is about its label prediction for a given node, and the more likely it is that the node would be selected for observation at the next iteration. Another uncertainty-based policy that is the state-of-the-art among Bayesian approaches is Bayesian Active Learning by Disagreement (BALD) which selects nodes that maximise the mutual information between label predictions and model posterior [22]. A number of alternatives to uncertainty-based policies exist in the AL literature, e.g., graph-based heuristics and Expected Error Reduction [37]. In this study we focus our attention primarily on policies that rely on graph-based heuristics and uncertainty-based policies that are adaptive (i.e. nodes to be observed are selected iteratively using information from previous observations); a summary of all policies considered in our experiments can be found in Table 1.

## 3 METHODOLOGY

### 3.1 Surrogate Model (CAR)

The Conditional Autoregressive (CAR) model is widely used in the small area estimation domain [5], where data consists of observations $\boldsymbol{y} = [y_1, y_2, \ldots, y_N]$ over a set of $N$ spatial units, which in the context of our study represent locations in a mobility network. The CAR model assumes that the value of a variable at a given node (or location) depends on the values at its neighbouring locations, with weights specified by a spatial adjacency matrix $\boldsymbol{A}$. For unweighted models, like the ones we work with in this paper, the adjacency matrix $\boldsymbol{A}$ is binary and captures the presence or absence of edges between corresponding nodes. The spatial random effect $\boldsymbol{f} = [f_1, f_2, \ldots, f_N]$ follows the multivariate normal prior with precision matrix $\boldsymbol{Q}$:

$$\boldsymbol{f} \sim \mathcal{N}(0, \boldsymbol{Q}^{-1}) \tag{1}$$

$$\boldsymbol{Q} = \tau(\boldsymbol{A} - \alpha\boldsymbol{D}) \tag{2}$$

The parameter $\alpha$ captures the amount of spatial correlation: if $\alpha = 0$, the model reduces to a set of independent errors at each location; and if $\alpha = 1$, the model reduces to the ICAR (intrinsic conditional autoregressive) model. In this study, we set $\alpha$ to a fixed value at 0.95 to clearly separate the tasks of spatial inference on graph from the task of optimisation test allocation. We use $\tau \sim \text{logNormal}(0, 0.1)$ as prior on the marginal precision.

CAR, as well as ICAR, are standard models in spatial statistics. Similar to Gaussian Processes (GPs), which are a standard choice

epiDAMIK 24, August 26, 2024, Barcelona, Spain.

Table 1. Summary of policies considered in this study. Abbreviation for each policy is shown in brackets following the policy name. For all policies, random tie-breaking is performed if and when there are multiple candidate nodes given equal preference according to a selection criterion.

| Allocation Policy | Policy Type | Brief Description |
|---|---|---|
| Least-Confidence (LC) [28] | Uncertainty-based Adaptive | Select the unlabelled node with predicted infection probability (posterior mean) that is closest to 0.5, indicating the least confidence in label prediction. |
| Node-Entropy (NE) [28] | | Select the unlabelled node with the highest entropy in its label prediction according to the surrogate model. It can be shown that NE always selects the same node as the policy LC at any iteration (see Appendix B. of the supplementary document [43]); as a result, only NE is considered hereafter. |
| Bayesian Active Learning by Disagreement (BALD) [22] | | Select the unlabelled node with the highest mutual information between label prediction and posterior from the surrogate model. |
| Local-Entropy (LE) *(our proposed policy)* | | Select the unlabelled node with the highest *Local-Entropy*, as defined by Equations (3)–(5), with $\lambda = 0$ (maximal exploration). |
| Degree-Centrality (DC) | Graph-based Non-Adaptive | Select the unlabelled node with the highest degree-centrality (most connections). |
| PageRank-Centrality (PC) | | Select the unlabelled node with the highest PageRank-centrality [7]. |
| Reactive-Infected (RI) | Benchmark Adaptive | Select at random an unlabelled node among immediate neighbours of nodes that are known to be infected from previous observations, if available; otherwise, sample randomly from remaining unlabelled nodes. |
| Random (RAND) | Benchmark Non-Adaptive | Select an unlabelled node at random. |

for surrogates over continuous space, CAR is the default model choice for modelling over a discrete set of areas. Future work should consider a wider range of surrogates, such as GPs on graphs when no knowledge about the spread of the disease is available; or mechanistic models, such as SIR and SEIR models, when the underlying mechanisms of the disease spread are well understood.

## 3.2 A Novel Policy: Selection by Local-Entropy (LE)

One potential drawback of uncertainty-based policies is that they can lead to a bias in favour of selecting nodes from regions with highly heterogeneous node labels. In the context of disease surveillance, this can be interpreted as an exploitation-exploration trade-off, where exploitation means the selection of nodes that lie along the boundaries between infected and uninfected regions (i.e. decision-boundaries), and exploration means the selection of nodes from less observed regions of the graph. Previous attempts to account for this trade-off have been made, particularly in the context of AL with GNN models, where exploration of less observed regions is encouraged by increasing the probability that a node is selected according to the number of unlabelled neighbours to which it is connected [31], or the degree to which the candidate node is representative of its unlabelled neighbours in feature space according to their node attributes [9].

With insights from these previous efforts, here we propose a novel policy which we refer to as Selection by *Local-Entropy (LE)*, whereby the informativeness of an unlabelled node is evaluated by taking into account not only the uncertainty in the predicted label of the candidate node itself, but also that of connected nodes. At a given iteration $r$, we define the Local Entropy of an unlabelled node

$v_k$ as a linear combination of the entropy of the label prediction for node $v_k$ itself denoted by $\Omega_{k,r}^{self}$, and the distance-weighted average entropy of the label predictions for surrounding nodes, denoted by $\Omega_{k,r}^{surr}$, as follows,

$$\Omega_{k,r} = \lambda \Omega_{k,r}^{self} + (1 - \lambda)\Omega_{k,r}^{surr} \tag{3}$$

with $\lambda \in [0, 1]$, and

$$\Omega_{k,r}^{self} = \mathrm{H}(v_k | \boldsymbol{D_r}) \tag{4}$$

$$\Omega_{k,r}^{surr} = \frac{\sum_{d=1}^{d_{max}} \sum_{v_i \in V(d,v_k)} \mathrm{H}(v_i | \boldsymbol{D_r})/d}{\sum_{d=1}^{d_{max}} \sum_{v_i \in V(d,v_k)} 1/d} \tag{5}$$

where $\mathrm{H}(v_i | \boldsymbol{D_r})$ is the entropy of the label prediction for node $v_i$, conditioned on the currently observed data $\boldsymbol{D_r} = \{(v_1, y_1), (v_2, y_2), \ldots, (v_n, y_n)\}$, and the entropy (for binary random variables) is defined as

$$\begin{aligned} \mathrm{H}(v_i | \boldsymbol{D_r}) = &- p(v_i | \boldsymbol{D_r}) \log p(v_i | \boldsymbol{D_r}) \\ &- [1 - p(v_i | \boldsymbol{D_r})] \log [1 - p(v_i | \boldsymbol{D_r})] \end{aligned} \tag{6}$$

Note the double summations in the expression for $\Omega_{k,r}^{surr}$ (Equation (5)), with the first summing over all neighbourhoods at different $d$-hop distances from candidate node $v_k$ for $d$ $in[1, d_{max}]$. Here, $d_{max}$ is an integer parameter whose value is bounded by the diameter of the graph, $d_G$, i.e. the greatest geodesic distance between any pair of nodes; it determines the cut-off in $d$-hop distance beyond which the observed label of a node is assumed to have a negligible effect on the label prediction of an unobserved node (radius of influence). The second summation sums over the entropy of the label prediction for all nodes in a given $d$-hop neighbourhood of the

candidate node $v_k$ (denoted by $V(d, v_k)$), weighted by the inverse of the geodesic distance, $d$.

Key insights that motivate the above definition of *Local Entropy* can be summarised as follows:

(1) The information that can be gained from the observation of a node is likely to be greater if it is in close proximity to other unlabelled nodes with highly uncertain label predictions.

(2) The influence that a new observation has on the label prediction of surrounding nodes decays with increasing $d$-hop distance. This, together with insight (1), motivates the definition of $\Omega_{k,r}^{surr}$, i.e. the sum of the entropy of the label prediction of all surrounding nodes (up to $d$-hop distance $d_{max}$) weighted by $1/d$, as a proxy measure of the total impact that the new observation is likely to have on the label predictions of surrounding nodes.

(3) This sum, as described in (2), is normalised by sum of the distance-weights across all $d$-hop neighbourhoods (up to $d$-hop distance $d_{max}$); this is to avoid the bias where centrally located nodes would have larger values of $\Omega_{k,r}^{surr}$, simply as a result of having more connections.

(4) The balance between exploitation and exploration, as described previously, can be fine-tuned by specifying different values of $\lambda$; in the case where $\lambda = 1$, we recover the uncertainty-based policy which performs node selection based on node-entropy alone.

Note that we set $d_{max}$ to the graph diameter, $d_G$, in all our following experiments; we leave the exploration of different values of $d_{max}$ for future work. We also set $\lambda = 0$ in all subsequent considerations of our proposed policy LE (i.e. maximal exploration). Results from a sensitivity analysis comparing the performance of LE at $\lambda = 0$, 0.5, and 1 on an aperiodic lattice graph can found in Appendix G. of the supplementary document [43].

## 4 EXPERIMENTS

We evaluate and compare the performance of the different policies (as summarised in Table 1) in three sets of experiments. In the first set of experiments, we consider an aperiodic lattice graph (with square-tiling), where each node has degree 4 except for those in the corners and at the edges of the lattice. To account for the inherent randomness in the stochastic SI process, we simulate 50 outbreaks realisations, with each outbreak terminating when at least 30% of the nodes become infected ($I/N = 0.3$). In the second set of experiments, we consider four different synthetic graphs, namely: 1) a periodic lattice graph (with square-tiling; each node has exactly four connections, unlike its aperiodic counterpart; Fig. 2a), 2) a random graph generated by the Barabási-Albert (BA) model, with each node having a minimum of two connections ($m = 2$) (Fig. 2b), 3) a random graph generated by the stochastic block (SB) model with low modularity (Fig. 2c), and 4) a random graph generated by the SB model with high modularity (Fig. 2d) (see Appendix E. of the supplementary document [43] for more details). For each graph, we again simulate 50 random outbreaks for each termination condition, i.e. when 10% ($I/N = 0.1$), 30% ($I/N = 0.3$), and 50% ($I/N = 0.5$) of the nodes become infected. Finally, in the third set of experiments, we consider graphs constructed from two empirical

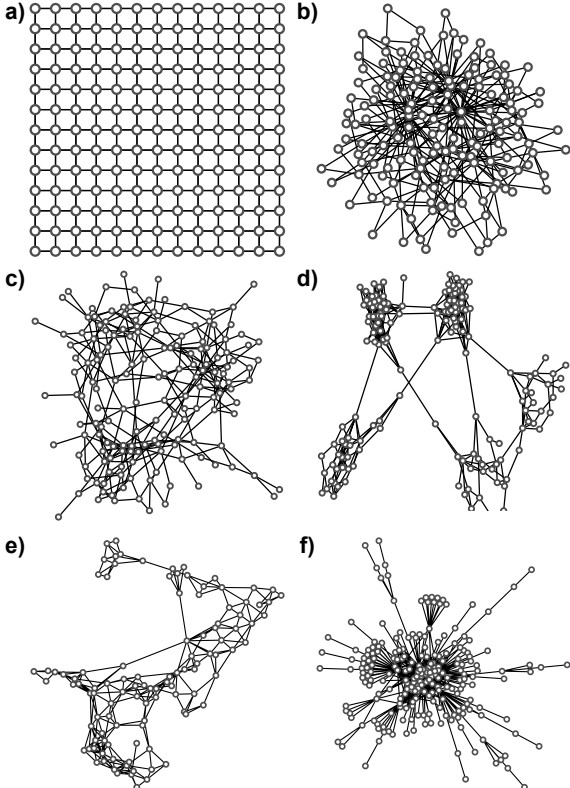

Fig. 2. Graphs considered in this study, including synthetic graphs ((a): a lattice graph with square-tiling; (b): a random graph generated by the Barabási-Albert model; (c): a random graph generated by the stochastic block model with low-modularity settings; (d): a random graph generated by the stochastic block model with high-modularity settings), and graphs constructed from empirical human mobility data ((e): within-country mobility data collected from mobile phone users in Italy, with thinning-threshold at 15%; (f): between-country air traffic data, with thinning-threshold at 5%).

human mobility datasets, namely: 1) aggregated mobility data from mobile phone trajectories collected in Italy at a provincial level in 2020 [35] (Fig. 2e), and 2) global air traffic data collected at a country level in 2020 [38] (Fig. 2f). See Appendix C. and D. of the supplementary document [43] for more descriptions of these two datasets, which are both openly available.

For each random outbreak realisation, 25 different nodes are randomly selected as the initial labelled node; at the beginning of each experiment, the infection status of the same selected node is made available to all agents to ensure a fair comparison between policies. This is done to account for any variability in policy performance resulting from different initial observations, stochasticity from the Markov chain Monte Carlo (MCMC) inference process and random tie-breaking when two or more candidate nodes are given equal preference by a policy according to its selection criterion.

In the experiment where we consider empirical human mobility data, an unweighted and undirected graph has to be constructed from each dataset by (i) first symmetrising the matrices representing

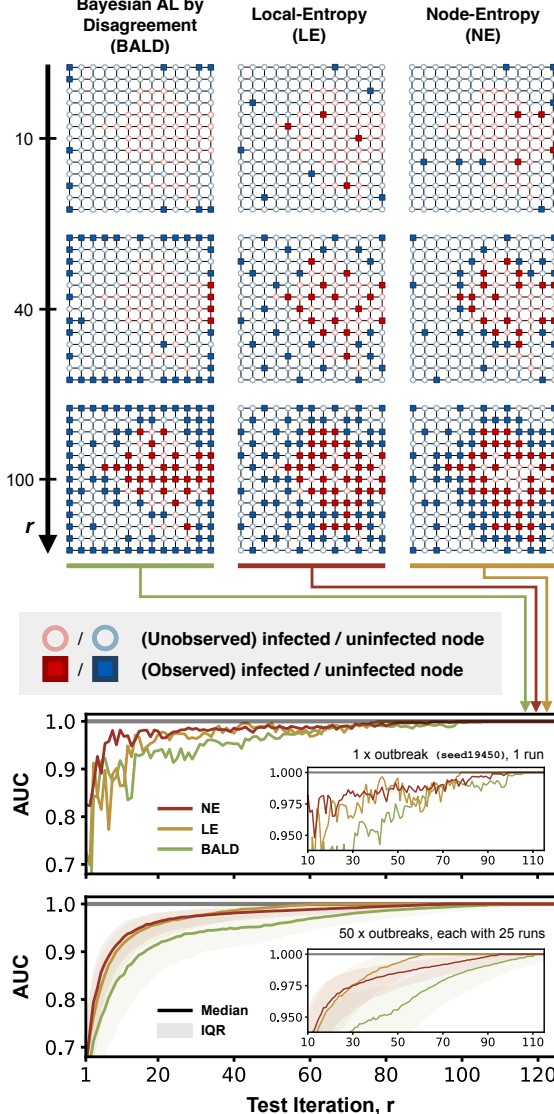

Fig. 3. Top panel shows the test allocation up to three different iterations ($r = 10$, $r = 40$ and $r = 100$) by three selected agents, each with a different designated (uncertainty-based) policy. Bottom panel shows the performance of the three selected agents (top), and the aggregated performance of the three policies (bottom), each summarised across 1,250 agents (50 outbreak realisations, each with 25 unique initial labelled nodes). In the bottom plot, shaded regions represent the interquartile range and the solid lines represent the median.

the mobility flows and ii) averaging them over the collection period, followed by (iii) removing edges with mobility flow below a certain threshold (also known as graph-thinning; see Appendix C. and D. of the supplementary document [43]). We also perform a series of sensitivity analyses to ensure that our results are robust to different thinning-thresholds (see Appendix J. and K. of the supplementary document [43]).

## 4.1 Test Budget and Performance Evaluation

The performance of each policy at a given iteration (or test budget) $r$ is evaluated by calculating the Area Under the Receiver Operating Characteristics Curve (AUC), based on the model predictions from the surrogate model conditioned on the observed data up to the current iteration, $D_r$; an AUC of 0.5 indicates no discriminative power and 1 indicates perfect predictions. Note that the AUC score is only evaluated for the set of nodes with unknown infection status at each iteration, as opposed to a held-out validation set.

Note that we also consider a policy referred to as Reactive-Infected (RI) designed to mimic the decisions of a policymaker whose aim is to identify as many infected locations as possible with the given resources, i.e. a "contact-tracing" approach. This policy provides a benchmark for the average test budget required to identify all infected nodes in a given outbreak scenario. It is therefore only at test iterations below this test budget that the objective of accurately predicting the presence or absence of a disease of interest may be considered relevant to public health decisions. In all following experiments, we compare the performance of the different policies only at test iterations up to this benchmark (median number of test iterations needed by RI to identify all infected nodes across all outbreak scenarios for a given graph); full results can be found in Appendix H. and I. of the supplementary document [43].

## 4.2 Disease Surveillance on an Aperiodic Lattice Graph

As a preliminary experiment to illustrate the differences between the uncertainty-based policies considered, we evaluate and compare their performance on an aperiodic lattice graph (with square-tiling). From the bottom plot in Fig. 3, we observe that LE on average performs better than both NE and BALD at small numbers of test iterations ($r < 30$). LE and NE show similar performance between $r = 30$ and $r = 50$; at $r > 50$, however, NE overtakes LE as the best performing policy with an AUC that rapidly approaches 1, while both LE and BALD struggle to attain a perfect AUC. This difference in performance between LE and NE can be understood in the context of the exploitation-exploration trade-off, as described previously: at small $r$, LE encourages an even allocation of tests across the graph (exploration), while NE favours regions with highly heterogeneous disease distributions (exploitation) (top panel in Fig. 3) - this results in a more rapid increase in model performance for LE as $r$ increases. At large $r$, however, the greater preference for exploitation by NE results in almost all of the nodes that lie along the decision-boundary being sampled - this results in an AUC that rapidly approaches 1. Although LE also shows a preferential selection of nodes close to the decision-boundary at large $r$, it does so at a much slower rate compared to NE.

BALD performs on average worse than both NE and LE across all test iterations. This is due to an apparent preferential selection of low-degree nodes (either in the corners or along the edges); only at $r > 40$ (at which point there are no remaining low-degree nodes to be observed) do we see a pattern of test allocation that roughly resembles that of NE. Briefly, this can be explained by noting that the second term in the selection metric for BALD [22] represents the average entropy of the posterior infection probabilities. For a given surveillance exercise, as the test iteration and therefore

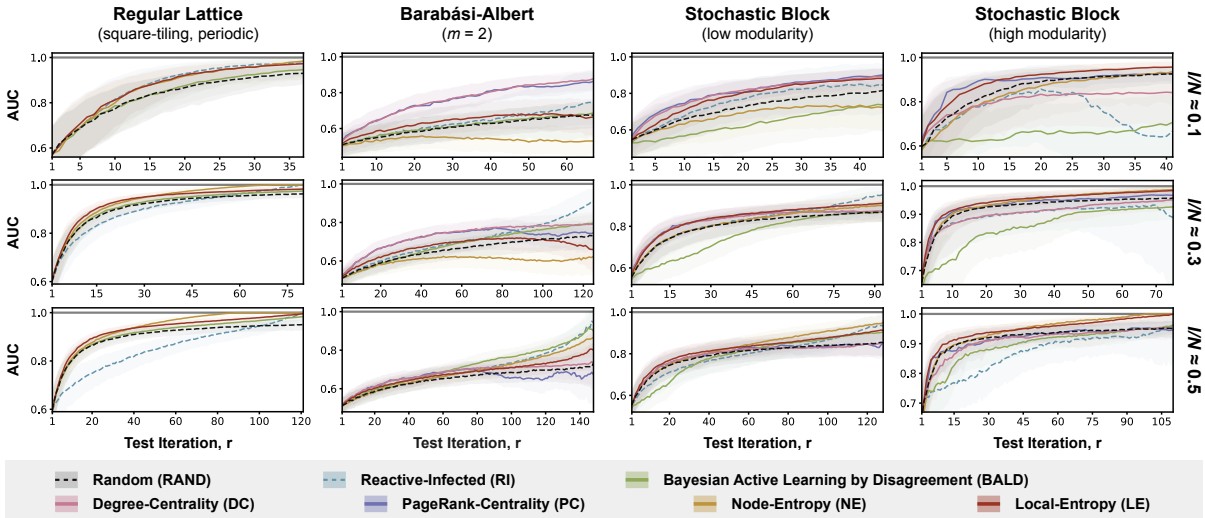

Fig. 4. Performance of policies considered in experiments with simulated disease distribution at different stages of outbreak progression (as indicated by labels on the right) and on different synthetic graph (as indicated by labels at the top). Shaded regions represent the interquartile range and the solid lines represent the median. Performance is only shown up to the median number of test iterations required for all infected nodes to be observed under the policy Reactive-Infected (RI). For corresponding numerical results, refer to Appendices L. to V. in the supplementary document [43].

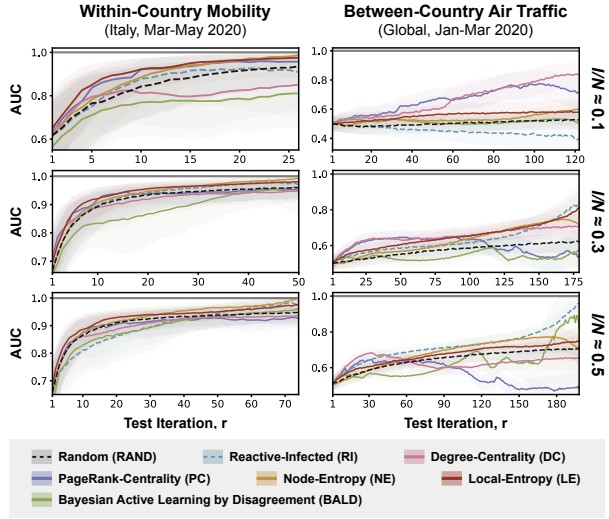

Fig. 5. Performance of policies considered in experiments with simulated disease distribution at different stages of outbreak progression (as indicated by labels on the right) and on two graphs constructed from empirical human mobility data (left: within-country mobility in Italy, 2020; right: between-country air traffic in 2020). Shaded regions represent the interquartile range and the solid lines represent the median. Performance is only shown up to the median number of test iterations required for all infected nodes to be observed under the policy Reactive-Infected (RI). For corresponding numerical results, refer to Appendices W. to AB. in the supplementary document [43].

number of observations increases, this term increases more quickly

for high-degree nodes compared to low-degree nodes, while the first term varies independently of node-degree. Overall, this results in the preferential selection of low-degree nodes, especially at small $r$ when the second term dominates over the first term.

### 4.3 Disease Surveillance on Synthetic Graphs

There are three key observations from our results presented in Fig. 4. First, all policies except for BALD and RI outperform random allocation (RAND) across most outbreak scenarios, especially at large $r$ when the performance of random allocation appears to only increase slowly with increasing $r$. Given the preferential selection of low-degree nodes by BALD as described in the previous section, it is not surprising that BALD only shows comparable performance in the periodic lattice graph which has no degree variation. Secondly, uncertainty-based policies (NE, BALD and LE) underperform substantially compared to graph-based policies (DC, PC) on the synthetic graph generated by the BA model (referred to as the BA-graph hereafter), with NE performing worse than RAND for $I/N = 0.1$, 0.3. This observation can be explained by considering a quantity known as infection-assortativity, $r_{\text{infection}}$, which in the context of disease distribution, is a measure of the tendency for two connected nodes to share the same infection status (see Appendix F. of the supplementary document [43]). Evaluating the average $r_{\text{infection}}$ across all 50 outbreak realisations on each graph shows that outbreaks on the BA-graph have on average the lowest $r_{\text{infection}}$ at 0.12 (compared to 0.89 for the periodic lattice graph, 0.45 and 0.55 for the graphs generated by the SB model (SB-graphs) with low and high modularity [32], respectively). A low (but positive) $r_{\text{infection}}$ indicates a weak tendency for two connected locations to share the same infection status, and therefore a low degree of homophily in the underlying disease distribution. This results in an overall poor

predictive performance from the surrogate model, which in turn limits the effectiveness of the uncertainty-based policies. In such cases, it may then be advantageous to consider node-centrality alone during node selection, especially at small $r$ when there is little data to inform model predictions. Note also that PC tends to perform better than DC - this is not unexpected given that nodes with the most connections are not necessarily the most central in a network.

Finally, we observe generally favourable performance from LE across most of the different outbreak scenarios considered on graphs with a high degree of structural order (unlike the BA-graph, as described), especially at small $r$. At larger $r$, however, we again observe superior performance from NE with AUCs that rapidly approach 1 - this can again be explained by the preference for exploitation over exploration by NE, which leads to the complete observation of the decision-boundary between infected and uninfected regions given a sufficient number of test iterations.

### 4.4    Disease Surveillance on Empirical Human Mobility Networks

From Fig. 2, it is clear that the two graphs constructed from empirical human mobility data ((e) and (f)) have markedly different structural properties. Graph-A, generated from aggregated mobility data derived from mobile phone trajectories in Italy at a provincial-level [35], shows distinct community structures with close resemblance to the SB-graphs as described in the previous section; whereas Graph-B, generated from the global air traffic data collected at a country-level [38], displays structural properties that are similar to those of the BA-graph, as consistent with previous studies which show the global air traffic network to have scale-free properties [11, 17] (e.g., both have a negative degree-assortativity (-0.25 and -0.27 for the BA-graph and the Graph-B, respectively; see Appendix F. of the supplementary document [43] for more details), indicating a hub-and-spoke as opposed to hub-and-hub structure [41]).

Indeed, from Fig. 5 we observe policy performances on Graph-A and Graph-B that are similar to those from experiments on the SB-graphs and BA-graph, respectively. Most notably for Graph-A, LE again shows rapid increases in model performance given small numbers of test iterations, only to be surpassed by NE at large $r$, as expected. For Graph-B, graph-based policies (DC, PC) outperform uncertainty-based policies especially at small $r$, again consistent with results from experiments on synthetic graphs. However, the superior performance of these graph-based policies only extends to larger values of $r$ if the outbreak under surveillance is at the early stages (i.e. $I/N = 0.1$); at later stages of outbreak progression, the performance of these policies in fact decreases with further increases in $r$. This counterintuitive observation can be explained by considering the changes in the distribution of the decision-boundary between the infected and uninfected regions in the graph during a transmission process. At the beginning of an outbreak, nodes that are centrally located are more likely to be infected early on due to their high degree of connectivity. This implies that most of the decision-boundary between infected and uninfected regions can be found close to the central nodes, thus explaining the superior performance of graph-based policies which preferentially selected nodes with high degree of centrality. As the outbreak progresses,

the decision-boundary shifts towards the periphery of the graph with the already infected central nodes acting as secondary hubs of the emerging pathogen. This results in a decrease in the performance of graph-based policies, as the central nodes continue to be targeted while the peripheral regions of the graph (where most heterogeneities in the disease distribution lie) remain largely unexplored. Note that a similar drop in the performance of PC (second column in Fig. 4) at large $r$ during later stages of outbreak progression ($I/N = 0.3$ and $I/N = 0.5$) can also be observed.

The same reasoning can also potentially explain the unexpected superior performance of graph-based policies and comparable performance of RI at small $r$, especially during the early stage of an outbreak (first row in see Fig. 4). More generally, provided that the number of infected nodes is sufficiently small and that they are confined to a small local region of the graph, any policy with which there is a high probability of selecting an infected node is likely to perform well compared to other policies, especially given a small number of test iterations.

## 5    DISCUSSION

In this study, we addressed the question of how a finite amount of testing resources should be allocated across a network in order to maximise the information gained about the underlying distribution of a disease of interest. By modelling the decision-making process as a node classification problem with AL, we evaluated and compared the performance of existing AL policies under different outbreak scenarios and on networks with different structural properties. We proposed a novel policy which, unlike most existing uncertainty-based policies, considers not only the uncertainty associated with the label prediction of a candidate node itself, but also the average level of uncertainty in the neighbourhood through a quantity named *Local-Entropy*.

Our results show that in general there is not a single optimal policy that performs best across all outbreak scenarios - instead, the performance of a given policy depends on both the test budget available and the geometry of the underlying disease distribution, which is in turn determined by the network structures and the stage of outbreak progression. For example, graph-based policies which target central nodes generally perform better than uncertainty-based policies when the underlying disease spread cannot be modelled with high accuracy and certainty. However, as a result of the non-iterative nature of these graph-based policies, their performance especially given a large test budget may be limited by their failure to identify and sampling in regions with highly heterogeneous disease distribution. Uncertainty-based policies are generally more effective when there are well-defined community structures in the network, with policies that encourage greater exploration early on often outperforming those that target nodes lying between infected and uninfected regions, especially when given a small test budget.

As future work, this framework can be extended to consider transmission models with greater complexities (e.g., SEIR models, spatially-explicit models) and more realistic mobility networks (e.g.,

directed and weighted graphs), with additional constraints to account for practical considerations in test deployments (e.g., observational noise and delay in test feedback). We hope that our approach can serve as a starting point for the development of more sophisticated surveillance strategies to inform globally coordinated responses to future infectious disease outbreaks.

## 6 DATA AND MATERIALS AVAILABILITY

Code and analysis files used in this work are openly accessible from GitHub at https://github.com/joetsui1994/osga.

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
