# OpenReview forum: "Optimal Disease Surveillance with Graph-Based Active Learning"
_KDD.org/2024/Workshop/epiDAMIK — KDD 2024 Workshop epiDAMIK_

### Official Review · Reviewer_gWvS · 2024-06-24
**This paper proposes a novel policy (Selection by Local-Entropy), to utilize active learning in the problem of disease surveilance. They pose the problem as the node classification (e.g., predicting the existance of a disease of interest). Though the evaluation scheme comes from a strong assumption, such as infection starts with a single infected node, spreads with disease model SI, on a simple (unweighted, undirected) graph, overall the paper is well written and with a good motivation.**

**Rating:** 4
**Confidence:** 4

**Review:**

Overall, the paper is well-written and explores active learning in disease surveillance. Their proposed method shows good performance over the baselines in most of the experiments conducted in their experiment setup.

Pros
- Clear motivation for the problem and clear problem setup.
- The writing is of good quality, and it is mostly clear to follow the logic.

Cons
- The experiment setup can be improved. Currently, by design, the ground truth underlying distribution is forced to be local (e.g., single source, simple graph, SI model). I suspect an algorithm that observes an infected node and then starts BFS from that observed infection would perform well in this experiment setup. I suggest authors evaluate an underlying disease distribution with multiple infection sources and possibly on a more realistic graph and disease modeling setup.
- The spread of infections varies for each disease and their type. Authors fail to discuss this in the paper. E.g., For which infectious diseases shall we expect your method to succeed or fail?
- How many tests are allocated at each iteration? Looks like this information is missing in the manuscript.
- How are the network structure related to the underlying disease distribution?
- Please justify the thresholds or numbers used in the experiments. E.g., why did you terminate the outbreak when at least 30% of the nodes become infected?
- How are the AUC computed, exactly? Does it only include ‘unobserved nodes’ or also include ‘observed nodes’? Please make this clear in the manuscript.

---

### Official Review · Reviewer_2BnQ · 2024-06-27
**Accept**

**Rating:** 5
**Confidence:** 3

**Review:**

Summary
This paper introduces a novel method in Active learning (AL) to detect and track the spread of disease in a static network. The AL approach is formulated as a node classification problem, in which the agent’s objective is to test whether the chosen node is infected or not based on existing knowledge. Disease dynamics on a network is modeled as an SI stochastic process. This paper introduces Local-Entropy (LE) to consider the uncertainty level of the chosen node based on the uncertainty level (entropy) of its neighbors. LE is compared to uncertainty-based baseline policies and graph-based policies. Results show that LE perform well in most cases, however, depending on the stage of the outbreak (early or late), graph-based policies may perform better.

Review:
Reasons why each policy falls short compared to others in each scenario are stated clearly. Paper easy to follow. Infection probabilities prediction make sense.
It makes more sense to think of the network as a social network of human rather than the network of location. If the nodes of the network were to be locations, then it should also have the population aspect of each node and base the stochastic SI process on that. I think it is unlikely that a whole location would be infected the same way human infect each other (example: https://www.medrxiv.org/content/10.1101/2020.03.13.20035386v1.full.pdf). To improve the method to make it more applicable, the authors can define a threshold of infection for each nodes base on the node’s population, and based on that threshold, determine whether the node is infected or not.

Another aspect to note is that the I(t)/S(t) ratio will increase over time, which could lead to poor performance from the entropy-based methods.

How much better does the policy perform when N is big (because too many parameters can cause problems for CAR model)?

Grammar/Figures: Figure 3 needs the legend for the graph. Nowhere in the paper mentions which color belongs to which policies. A grammar error/extraneous word in section 3.1

---

### Official Review · Reviewer_PnKF · 2024-06-28
**Graph-based active learning to address limited surveillance resource distribution**

**Rating:** 5
**Confidence:** 4

**Review:**

The authors propose a method for optimizing surveillance resource allocation with a limited budget via iterative graph-based active learning. Their newly proposed active learning policy, "local entropy", extends node uncertainty by a term considering the uncertainty in neighboring nodes. The proposed policy is then systematically evaluated using synthetic and real-world mobility data.
### Quality
- the shown experiments are thorough and detailed with reasonable parameter choices
- while the experiments are systematic and well-structured, I would like to see an evaluation for local entropy at least one more $\lambda$ such as $\lambda=0.5$ to have a better understanding of the effects of a balanced exploration/exploitation trade-off
### Clarity
- The manuscript is well-written and well-structured
- The scope of the research addressed in this manuscript is clearly stated throughout the manuscript
- I appreciate the summary of key insights that motivate the proposed *local entropy*
- In section 4.3, the authors mention that "all policies except for BALD and RI outperform random allocation (RAND) across all outbreak scenarios, especially at large r". This statement seems to be too general considering the performance of NE for the BA-graph in the $I/N=0.1$ and $I/N=0.3$ scenarios and PC in the $I/N=0.5$ scenario and needs clarification
- Further, while the decreasing performance of PC for later stages is addressed for the synthetic experiments, this is only done towards the end of section 4.4. While I understand that this helps with the flow of the argument, mentioning the observation already in section 4.3 when the topic is initially mentioned might be helpful and provide context for the statement that "uncertainty-based policies (NE, BALD and LE) underperform substantially compared to graph-based policies (DC, PC)" and later "PC tends to perform better than DC"
- minor: in section 4.3, BA is used as an abbreviation for Barabási-Albert without a previous introduction
-  I would appreciate references to the different elements of Fig.2 directly from the text in section 4.
- Figure 3 the colors of the circles are somewhat challenging to see and the plot size should be increased
- In Figures 4+5, the shades representing the IQR make the plots even more challenging to read, considering the already small size
### Originality and Significance
- The authors build on a strong foundation of existing literature in the field of graph-based active learning but extend this to the field of resource allocation in disease surveillance, which is, to my knowledge, novel
- While the proposed uncertainty-based active learning policy does not outperform all
### Pros
- strong and clear motivation for the systematic distribution of surveillance resources
- reasonable assumptions for the evaluated scenario - even more realistic scenarios with relaxed assumptions promised in future work
### Cons
- I highlighted some potential improvements in the Quality and Clarity sections, mostly related to the presentation of the content already in the manuscript.
- My final concern is concerning the results being only deduced visually. While this makes much sense, the work could be further improved by including appropriate qualitative results.

---

### Official Review · Reviewer_3wZF · 2024-06-29
**Clearly written paper on graph active learning with a novel method for exploration, but the evaluation and the implications should be made stronger.**

**Rating:** 4
**Confidence:** 4

**Review:**

**Summary**:
The paper studies iterative node exploration for networks in the context of a disease outbreak. It models the disease by a CAR model, and assumes a static distribution after the initial outbreak. It then compares different test site allocation methods i.e. which nodes in the graph to test for infections and compares them across various random/real world graphs with the metric being AUC.  It concludes by stating that there isn't one single optimal test allocation policy across different networks and budgets.

**Strengths**:
- The paper is well written, self contained and runs experiments across various networks, and different test allocation rules
- The motivation and explanation for the choice of Local-Entropy is clear.

**Weaknesses**:
- "budget" is mentioned couple of times in the paper, but where is it explicitly accounted for in the node exploration rules?
- In your experiments is there some configuration of $\lambda, d$ for which LE does better than the other rules, this kind of result would strengthen the use case for the novel LE metric.
- Minor presentation issue: The plots for AUC across various allocation rules look good, but it's hard to make out when a method is better than the other. Perhaps a supplementary table with rows representing the methods and columns representing the time step, with bold text for the better rule at a given time step would be good to have.